# Functional Trait Diversity Shapes the Biomass in the Dam-Induced Riparian Zone

**Yanfeng Wang [1,2], Shengjun Wu [1], Ying Liu [1,\*], Xiaohong Li [1,2] and Jing Zhang [1,2]**

[1] The Three Gorges Institute of Ecological Environment, Chongqing Institute of Green and Intelligent Technology, Chinese Academy of Sciences, Chongqing 400714, China; wangyanfeng@cigit.ac.cn (Y.W.); wsj@cigit.ac.cn (S.W.); lixiaohong19@mails.ucas.ac.cn (X.L.); zhangjing@cigit.ac.cn (J.Z.)

[2] University of Chinese Academy of Sciences, Beijing 100049, China

\* Correspondence: liuying@cigit.ac.cn

**Abstract:** The construction of dams has caused a serious decline in riparian ecosystem functioning and associated services. It is crucial to assess the response of riparian plant communities to flooding stress for their conservation. Functional traits composition, functional diversity, and species diversity are commonly used to investigate the effect of abiotic stress on ecosystem functioning and services (i.e., biomass). Yet, how the functional traits respond to the flooding stress along a dam-induced riparian habitat remains unclear, and how biodiversity affects biomass still exists controversy. Accordingly, this study investigated the response strategies of functional traits subjected to the flooding stress and its correlation with aboveground biomass (AGB) in the water level fluctuation zone (WLFZ) of the Three Gorges Reservoir (TGR). We found that leaf traits and root traits showed a significant response to elevation, whereas they demonstrated different distribution patterns. Leaf traits showed acquisitive-conservative-acquisitive resource strategies along the flooding stress, while root traits shifted from species conservative resource to acquisitive resource strategies. AGB was found to be positively related to the community weighted mean (CWM) trait values for leaf dry matter content (LDMC) and negatively related to specific leaf area (SLA), but the AGB showed no relationship with the root traits. AGB accumulated greatly in the intermediate species diversity, and we also found a significant relationship between functional diversity and biomass within threshold values. Additionally, Rao's exerted the most significant influence on the biomass, suggesting that the functional diversity index is a better indicator of biomass variation. The results obtained only partly supported the "mass ratio hypothesis" in leaf traits and mainly supported the "niche complementarity hypothesis", which suggested that these two theories are not mutually exclusive at the early stage of vegetation community succession with an unstable community structure in dam-regulated riparian zones.

**Keywords:** biodiversity-ecosystem functioning; flooding stress; water fluctuation; reservoir riparian zone

## 1. Introduction

Plant functional traits are commonly used as indicators of species ecological functions, which can better understand resource use strategies and species adaptations along environmental gradients [1,2]. The plant economics spectrum (PES) describes the trade-offs between resource acquisition and conservation of plant functional traits [3–5]. A suite of correlated plant functional traits, such as low-density tissues, high metabolic capacity, and high nutrient uptake rate, are commonly associated with resource acquisition strategies of fast-growing species, whereas resource conservation of slow-growing species has the opposite traits [6,7]. Although the traits along the PES are related to resource use, plants also need to adapt to abiotic stress and show specific traits to survive and reproduce successfully [8]. For example, aquatic plants exhibit higher specific leaf area (SLA) to increase gas exchange and ensure optimal photosynthesis during submerged [9–11], whereas, in salt marshes, plants have lower SLA and leaf dry matter content (LDMC) to reduce evapotranspiration and maintain osmotic balance to cope with salinity stress [12]. However, the

plant strategies or adaptations to a range of environmental conditions along the riparian habitats, especially those distributed at the dam-induced riparian area governed by the periodic terrestrial-to-aquatic alternation, remains unclear. Therefore, understanding the adaptive strategies of riparian species to flooding stress along a dam-induced reservoir is beneficial to conserve and restore riparian ecosystems.

Functional traits composition, functional diversity, and species diversity could be used to investigate the effect of abiotic stress on ecosystem functioning and services (i.e., biomass) [13,14]. Understanding the relationship between biodiversity and aboveground biomass (AGB) is of great significance for managing carbon storage in AGB and slowing down the increasing atmospheric $CO^2$ concentration [15,16], and thus, more attention has been drawn to explore its relationship. However, the effect of biodiversity on biomass still exists in controversy. The mass ratio hypothesis states that ecosystem processes are driven by community weighted mean (CWM) of trait values [17,18], which means the effect of traits on biomass depends on the dominant species in plant communities and plants with a single resource use strategy would have high AGB. Alternatively, the niche complementarity hypothesis proposes that communities with more diverse functional traits are prone to have high AGB due to complementary resource use among species [19,20]. The differences in functional traits among species maximize the diversity of resource strategies, and the functioning of the ecosystem is improved by niche partition with less overlap of niches along the resource axes [21]. In addition, some studies suggested that these two theories are not mutually exclusive [16,22] or even only partly supported the mass ratio hypothesis or niche complementarity hypothesis [14]. In the case of relationships between diversity and AGB [23–25], fewer studies covered how functional trait diversity and composition influence AGB in dam-regulated riparian habitats. Therefore, it is necessary to conduct more studies to explore the main drivers of AGB. The AGB is mainly affected by various abiotic and biotic factors such as species richness, CWM, functional diversity, and elevation [26–28], which may be helpful for the government to adopt different measures to restore the vulnerable riparian ecosystem. However, it remains a challenge in quantifying the relative importance of determinants of the AGB in riparian habitats.

With the construction and impoundment of Three Gorges Dam (TGD) in 2010, a large riparian zone has been formed in the upper reaches of the Yangtze River [29]. Its original hydrological regime was restructured due to the operation of the dam; that is, the water level fluctuates between 145 and 175 m above sea level [30]. Thus, the riparian fluctuation range is as high as 30 m, forming the biggest water level fluctuating zone (WLFZ) of approximately 350 km$^2$ in China [31]. The flooding time remains more than 8 months every year. At the beginning of September, the water level begins to rise, and it takes two months to reach the maximum water level at 175 m above sea level, where it lasts 2 months. Then, from January to May, its water level gradually drops to a minimum of 145 m above sea level [32]. The abiotic stress gradients caused by the periodic terrestrial-to-aquatic alternation make the WLFZ of the Three Gorges Reservoir (TGR) an ideal system to investigate the effect of biodiversity on AGB. Therefore, this study aimed to address three questions: (1) How are CWM traits associated with elevation? (2) Which hypothesis best explain variation in AGB? Mass ratio hypothesis, niche complementarity hypothesis or other hypothesis? (3) Which of the factors, CWM, species diversity, functional diversity, and elevation plays the greatest influence on AGB?

## 2. Materials and Methods

### 2.1. Study Area

The study area, covering approximately 55.47 km$^2$, is located in the upper-mid section of a primary tributary (named Pengxi River) of the Yangtze River and is regulated by the Three Gorges Dam (Figure 1). The water level in this region falls to 150 m in summer and rises to 175 m in winter because the Hanfeng Lake dam was constructed on the Pengxi River, and only a section of the river is fully controlled by the Three Gorges Dam [33]. This region is characterized by a humid subtropical monsoon climate with a mean annual air

temperature that varies from 15 to 18 °C and a mean annual precipitation of 1200 mm [29]. About 80% of annual falling occurs from April to October [34]. The amount of rainfall in the study area is around 150–170 mm during September of 2019 and 2020. The soil types are classified as Regosols and Anthrosols (FAO Taxonomy) [35]. The type of vegetation is herbaceous, which is dominated by *Cynodon dactylon*, *Xanthium sibiricum*, *Cyperus rotundus*, *Alternanthera philoxeroides,* and *Setaria viridis* [36].

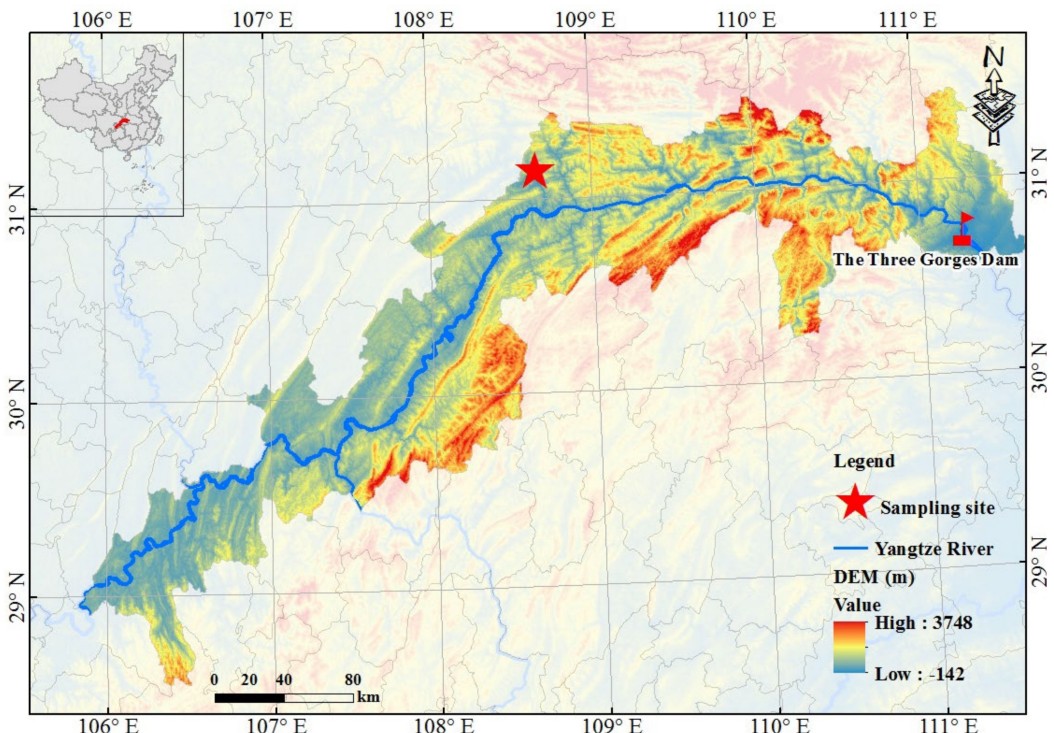

**Figure 1.** The location of sampling sites and DEM in the WFLZ of the TGR.

## 2.2. Vegetation Survey

The field study was carried out in September of 2019 and 2020 when the water level of the study area came to the lowest level, around 150 m. Four transects were selected for investigation, and three to five quadrats (1 × 1 m) were chosen along each transect at 1–5 m intervals (from 150 to 175 m). In the end, a total of 59 quadrats were investigated. In each quadrat, coverage, abundance, and life form of all rooted species appeared were recorded. Plant coverage was estimated based on the percentage of the plot surface area of each species present. GPS was applied to locate the geographic position and elevation of each quadrat with 1 m accuracy. Plant nomenclature follows the Flora of China [37]. One-fourth of aboveground plants in a quadrat (0.5 × 0.5 m) were clipped and then dried at 80 °C for 48 h in the lab.

## 2.3. Traits Measuring

Following the standard trait collection protocols [38], 1 to 5 individuals of each species in the quadrats were sampled. For each individual, we collected at least three leaves and their roots. Fresh roots were carefully removing the surface soil, washed by hand, and drained surface moisture with paper towel in the lab. The leaf area and the root traits length, volume, and average diameter were measured on fresh roots using the WinRhizo$^{TM}$ scanner-based system. Fresh leaves and roots were oven-dried at 80 °C for 48 h to measure dry mass. We analyzed the functional diversity of plant communities based on six plant traits, including LDMC, SLA, RDMC, SRL, RTD, and RAD [39] (Table 1).

**Table 1.** Functional traits measured in this study.

| Traits | Abbreviation | Description | Unit |
|---|---|---|---|
| Leaf dry matter content | LDMC | The ratio of the leaf dry mass to fresh mass | g/g |
| Specific leaf area | SLA | The ratio of the leaf area to dry mass | $cm^2/g$ |
| Root dry matter content | RDMC | The ratio of the root dry mass to fresh mass | g/g |
| Specific root length | SLR | The ratio of root length to root dry mass | cm/g |
| Root tissue density | RTD | Root dry mass per root volume | $g/cm^3$ |
| Root average diameter | RAD | Average diameter of all roots | cm |

*2.4. Data Analysis*

The community weighted mean trait values (CWM) of each trait in every sampled quadrat were calculated as [40]:

$$\text{CWM} = \sum_{i=1}^{n} p_i \, \text{trait}_i$$

where $n$ is the number of species in the quadrat, $p_i$ is the relative coverage of species $i$, $\text{trait}_i$ is the trait value of species $i$.

The taxonomic diversity, including Shannon–Winner diversity and Simpson diversity, were calculated by the "vegan" package in R, and functional diversity metrics including functional dispersion, functional evenness, functional divergence, and Rao's index were calculated by using the "dbFD" package. The relationships between CWM/biomass and the elevation and how AGB related to CWM, taxonomic diversity, and functional diversity were analyzed with the linear model and quadratic regression, and goodness-of-fit was evaluated by *p*-value ($p < 0.05$). A generalized additive model (GAM) was used to test the relative effect of taxonomic diversity, CWM, functional diversity, and elevation on AGB [41]. The original formulation of this model is

$$Y = \alpha + \sum_{i}^{n} f_j(x_i) + \varepsilon$$

where Y is AGB; $x_i$ is explanation variables (taxonomic diversity, CWM, functional diversity, and elevation of each survey plot); $\alpha$ is formulation intercept; $\varepsilon$ is residual; $f_j$ is smoothing function. The best GAM was obtained through a stepwise procedure by choosing a significant p-value for each variable as follows

$$Log(AGB + 1) \; = \; s(LDMC) \; + s(SLA) \; + \; s\left(Rao's\right) \, s(elevation) \; + \varepsilon$$

where $AGB + 1$, logarithmically transformation was taken on $AGB$, and 1 was added to avoid responsive variables being 0. s($LDMC$), the effect of $LDMC$ of CWM; s($SLA$), the effect of $SLA$ of CWM; s($Rao's$), the effect of Rao's of functional diversity; s($elevation$), the effect of elevation. $\varepsilon$, model error followed the Gaussian distribution.

Akaike information criterion (AIC) was applied to examine the fitness of the model after adding variables to the model. Generalized cross-validation (GCV) was used to assess predictive variables. The smaller the AIC value and GCV is, the better the model fitting effect [42]. The GAMs were constructed in R with the "mgcv" package. The significance of a predictive variable in the model and the nonlinear contribution of variables to nonparametric results evaluated by F-test and Chi-test, respectively. All analyses were conducted using R (Version 3.6.2).

**3. Results**

*3.1. Community-Level Trait along the Elevation*

The CWM of the six measured plant functional traits varied along the elevation. LDMC showed a convex relationship with the elevation, whereas SLA showed a concave relationship (Figure 2a,b). For root traits, RDMC and RAD displayed a linear increase with elevation, whereas RTD showed an opposite pattern (Figure 2c,d,f). In addition, the

CWM of SRL did not depend on elevation (Figure 2e). AGB was the highest in the middle elevation and showed a hump-shaped relationship with the elevation, similar to LDMC and contrary to SLA (Figure 3).

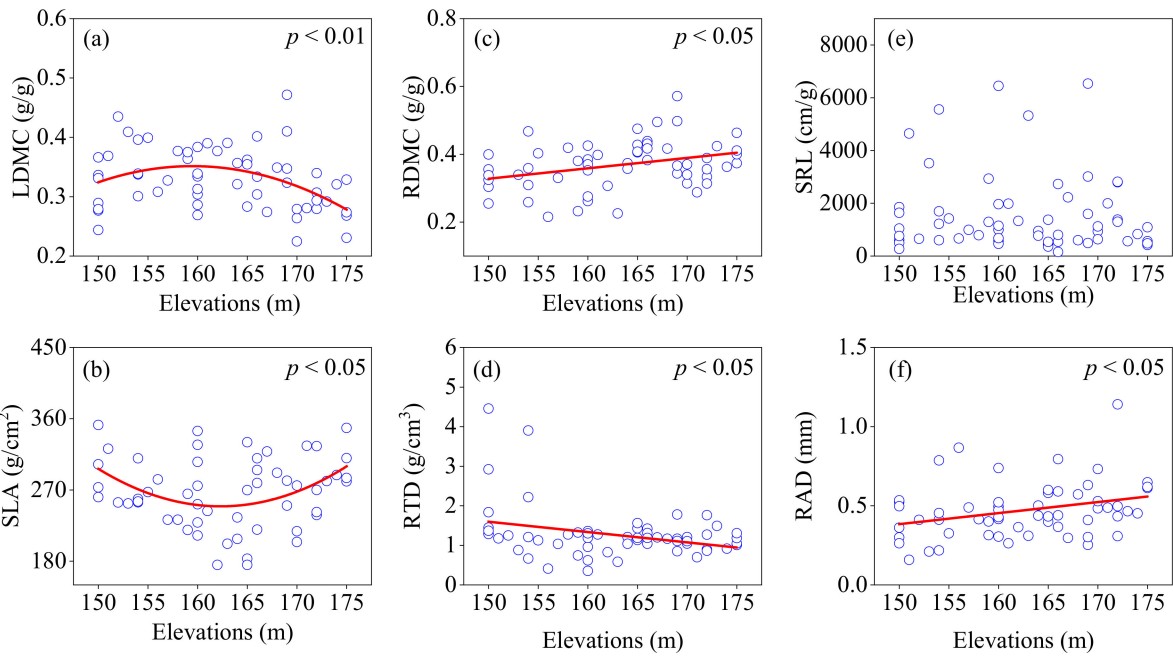

**Figure 2.** Effect of elevations on the community weighted mean (CWM) traits values of leaf dry matter content (LDMC) (**a**), specific leaf area (SLA) (**b**), root dry matter content (RDMC) (**c**), root tissue density (RTD) (**d**), specific root length (SRL) (**e**), and root average diameter (RAD) (**f**).

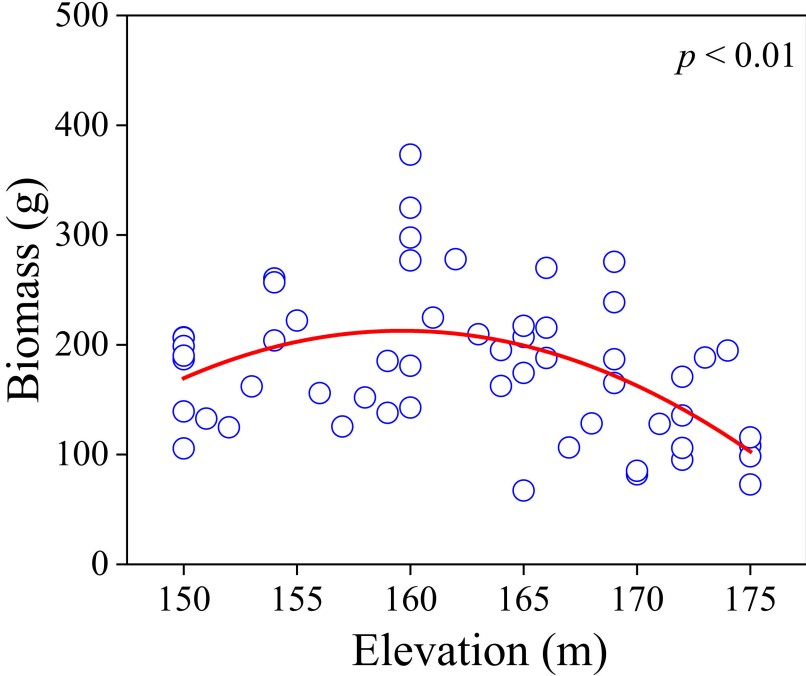

**Figure 3.** Effect of elevations on the aboveground biomass (AGB).

### 3.2. Relationships between AGB and Taxonomic Diversity

AGB was unrelated to species richness, whereas it was linked to a nonlinear relationship with Shannon diversity and Simpson diversity; that is, AGB increased and then decreased with increasing Shannon diversity and Simpson diversity (Figure 4a–c). A higher AGB value was associated with a Shannon's value of 2–4 and Simpson's value of 0.3–0.65.

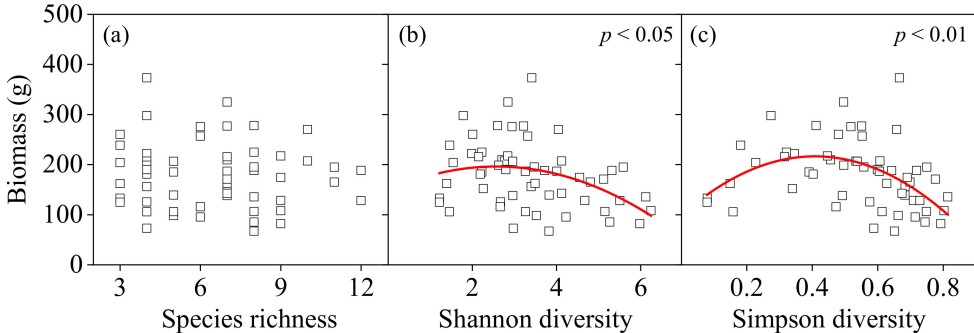

**Figure 4.** Relationships between aboveground biomass (AGB) and taxonomic diversity metrics (**a–c**).

### 3.3. Relationships between AGB and CWM Traits

AGB was found to be positively related to the CWM trait values for LDMC and negatively related to CWM for SLA (Figure 5a,b). However, the biomass showed no relationship with the root traits (Figure 5c–f).

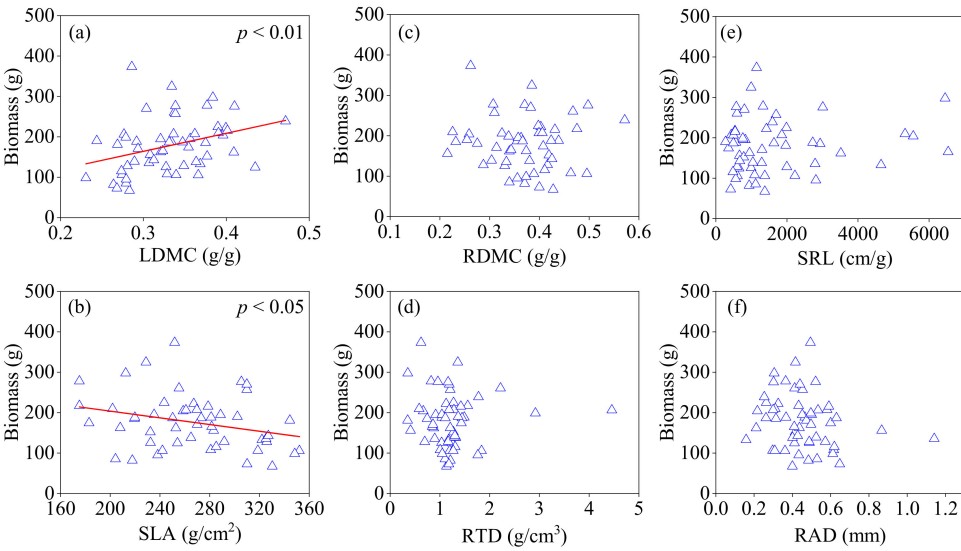

**Figure 5.** Relationships between aboveground biomass (AGB) and community weighted mean (CWM) trait values (**a–f**).

### 3.4. Relationships between AGB and Functional Diversity

Relationships between AGB and functional diversity were significant for functional dispersion, functional divergence, and Rao's. Functional dispersion showed a hump-shaped relationship with biomass (Figure 6a). The biomass was the highest when the functional dispersion value was 1.2, functional divergence value was 0.92, and Rao's value was 1.9 (Figure 6a,b,d).

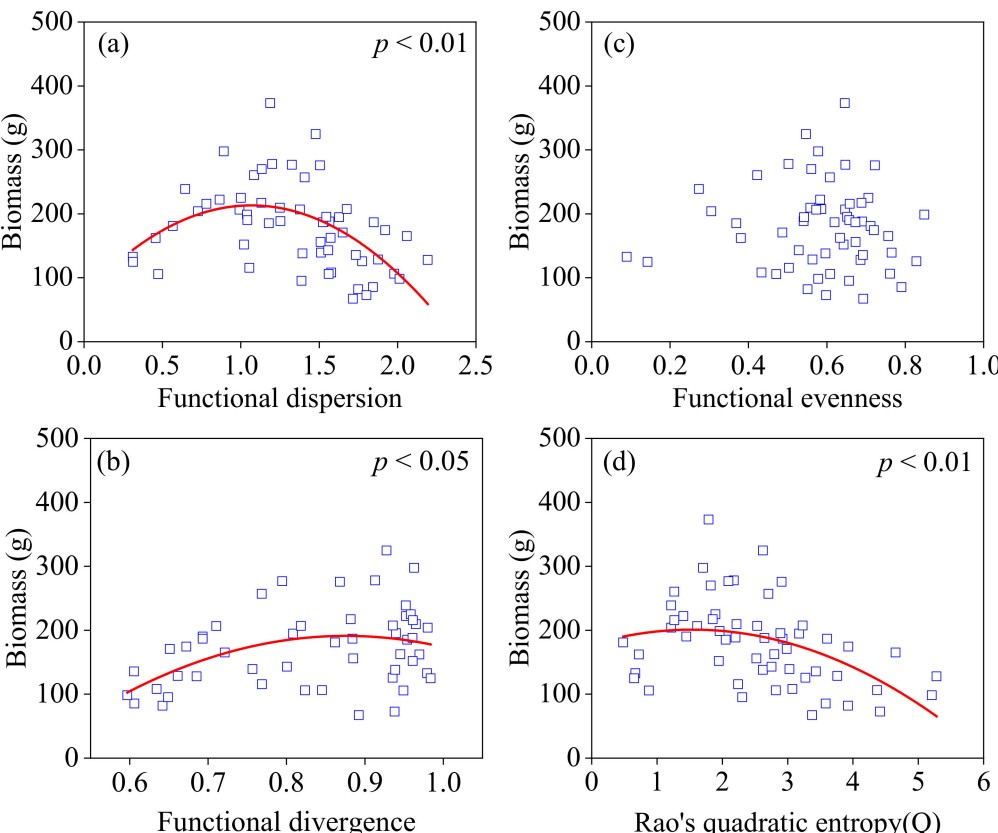

**Figure 6.** Relationships between aboveground biomass (AGB) and functional diversity metrics (**a–d**).

*3.5. Contributions of CWM Traits, Taxonomic Diversity, Functional Diversity, and Elevation on AGB*

GAM was used to explore the relative impact of variables on AGB. The best predictive GAM was the combination of Rao's, elevation, SLA, and LDMC. Their contribution to AGB was 65.9%, with $R^2$ = 0.56 (Table 2). Rao's played the strongest influence on AGB, with a relative contribution of 32.5%, followed by elevation, SLA, and LDMC, with the relative contribution being 15.4%, 10.1%, and 7.9%, respectively (Table 3).

**Table 2.** Deviance analysis for the general additive models (GAMs) fitted to the biomass.

| Model Factors | AIC Value | GCV Value | Adjusted $R^2$ Value | Cumulative Deviance Explained (%) |
|---|---|---|---|---|
| Log(AGB + 1) = s(SLA) | 48.72 | 0.15 | 0.07 | 10.1 |
| Log(AGB + 1) = s(SLA) + s(LDMC) | 45.97 | 0.14 | 0.14 | 18.0 |
| Log(AGB + 1) = s(SLA) + s(LDMC) + s(Rao's) | 28.63 | 0.10 | 0.43 | 50.5 |
| Log(AGB + 1) = s(SLA) + s(LDMC) + s(Rao's) + s(elevation) | 19.83 | 0.09 | 0.56 | 65.9 |

AIC, Akaike information criterion; GCV, generalized cross-validation; SLA, specific leaf area; LDMC, leaf dry matter content.

**Table 3.** Contributions of Rao's, elevation, SLA, and LDMC on biomass in GAMs.

| Variables | Contributions (%) | F-Test | Chi-Square Test |
|---|---|---|---|
| Rao's | 32.5 | 0.03 | 0.00 |
| Elevation | 15.4 | 0.04 | 0.00 |
| SLA | 10.1 | 0.05 | 0.00 |
| LDMC | 7.9 | 0.05 | 0.00 |

SLA, specific leaf area; LDMC, leaf dry matter content.

## 4. Discussion

### 4.1. Elevation Patterns of Community-Level Trait Variation

The leaf and root traits showed a significant response to elevation, whereas they showed different distribution patterns. The CWM values of SLA and LDMC in response to the elevation gradient showed acquisitive-conservative-acquisitive resource strategies, whereas the CWM of RTD suggested that species in our study area shifted conservative resource to acquisitive resource strategies, which is not consistent with the leaf economic spectrum, but root economic spectrum [43]. This result indicates that riparian plants may adopt specific adaptive strategies to cope with anti-seasonal flooding stress. For leaf traits, low SLA and high LDMC appeared in middle elevation, whereas high SLA and low LDMC appeared in the low and high elevation, possibly due to SLA and LDMC response differently to environmental factors along the elevation [44]. At low elevation, the short landing time in the WLFZ of TGR from June to September every year [45,46], leading to herbs enlarging SLA and reducing LDMC to grow fast and complete their life cycle quickly. At high elevation, more species colonize easily with the stress decreased [47], and thus herbs might increase their SLA to cope with the strong interspecific competition of light and water. However, species in the middle elevation showed opposite pattern in leaf traits. That finding may be ascribed to the intermediate stress hypothesis. Species diversity within habitats will be the highest at moderate levels of stress [48], which leads to nutrition limitation, and thus, plants may allocate resources preferentially to storge and defense in the middle elevation. Moreover, plants that exhibit lower SLA and higher LDMC could conserve more energy and escape from physiological nutrient stress [49]. Therefore, the CWM of SLA showed lower but LDMC higher in the middle elevation.

Root traits are multidimensional. The CWM of RTD decreased with increasing elevation, while the CWM of RDMC and RAD demonstrated opposite pattern. This is possibly because of severe flooding stress caused by long durations and deep depths in the WLFZ of the TGR, making only perennial clonal plants (e.g., *Cynodon dactylon*) with robust clonal growth and stolons between ramets and partly annual plants surviving at low elevation [50,51]. Perennial clonal plants are characterized by denser and thin roots, and thus, higher RTD and lower RAD and RDMC were found at low elevation. On the contrary, annual plants dominate at high elevations, and their roots are less dense and thicker. Moreover, annual plants were drowned and decomposition by flooding in winter, while perennial plants at the low elevation still live with strong adaptability to oxygen deficit, osmotic stress, and oxidative stress [30]. Therefore, lower RTD and higher RAD and LDMC appeared at high elevation.

### 4.2. The Effect of CWM on Biomass

As proposed by the "biomass ratio hypothesis", CWM trait values were linked to biomass production; that is, biomass production is determined by the dominant species in communities [16]. The results obtained in our study showed that the CWM of leaf traits (LDMC and SLA) was significantly related to AGB, whereas the CWM of root traits does not drive AGB significantly. On the one hand, the above- and underground traits of plants may undergo independent adaptive evolution under periodic flooding [52,53]. The differences of above- and underground physiological processes may weaken the relationship between AGB and root traits [54]. On the other hand, AGB is more easily affected by leaf traits than root traits, possibly because root traits would be encountered by more complex abiotic

and biotic selective pressures, and such constraints to root traits do not directly reflect on AGB [55,56]. For example, the spatial heterogeneity of soil nutrient availability and mycorrhizal fungi presents various constraints to root trait variation [57]. Hence, root traits could influence AGB indirectly through multiple mechanisms.

The CWM of LDMC was positively related to AGB, but SLA negatively, indicating that AGB can be affected by the dominant species with higher LDMC and lower SLA in the riparian zone. Plants subjected to prolonged flooding can undergo a "physiological drought" because oxygen deficiency in the submerged root environment prevents water uptake and nutrient absorption by plants [58]. A decrease in SLA and an increase in LDMC might reduce the water loss and conserve efficient nutrients in plant tissues under soil waterlogging [44], which helps the plant community accumulate biomass effectively.

*4.3. The Effect of Functional Diversity on Biomass*

In our study, the effect of taxonomic diversity and functional diversity on AGB revealed that AGB peaks at intermediate levels of diversity, which is in line with the intermediate stress hypothesis. One possible explanation is that the use of available limiting resources can be maximum and saturate due to the niche difference and facilitative interactions between species [59]. Therefore, biomass accumulated greatly in the intermediate species diversity. In addition, we also found a significant relationship between functional diversity and AGB within threshold values. Functional dispersion is often used as a surrogate for niche complementarity in resource exploitation [60]. Functional divergence indicates the degree of resource differentiation, and Rao's indicates the degree of niche overlap of species [61]. Therefore, these three functional diversity indices indicated that the degree of resource use depends on niche complementarity and niche overlap of species in communities. We found that the effects of niche complementarity favorable to AGB may indeed dominate up to functional dispersion values of 1.2, functional divergence values of 0.92, and Rao's values of 1.9. This is possible because community assembly is mainly dominated by environmental filters at low elevation and interspecific competition at high elevation [30,32,62]. When environmental filters select individuals based on their tolerance to resource availability, this leads to a restriction in the range of the amplitude of functional trait values with different functions and a convergent distribution with low dispersion around the functional trait mean [63]. In contrast, when interspecific competition increases with decreasing flooding stress, it may lead to an increase in dispersion or divergence of functional trait values [64]. Therefore, functional diversity values indicate AGB with a dilution effect across the WLFZ of the TGR. Additionally, Rao's exerted the most significant influence on the biomass, suggesting that the functional diversity index is a better indicator of biomass variation. The functional diversity index contains more information about the community structure, especially Rao's [65]. Therefore, functional diversity is more important for the AGB than taxonomic diversity and CWM traits.

## 5. Conclusions

This study evaluated the CWM traits and AGB response to flooding stress and the relationship between varied metrics of biodiversity (CWM traits, species diversity, and functional diversity) and AGB in the WLFZ of TGR. The results indicated that leaf traits showed acquisitive-conservative-acquisitive resource strategies along the flooding stress, while root traits shifted from species conservative resource to acquisitive resource strategies. AGB was the highest in the middle elevation, which is in line with the intermediate stress hypothesis. Moreover, AGB was positively related to the CWM trait values for LDMC and negatively related to SLA, but the AGB showed no relationship with the root traits. AGB accumulated greatly in the intermediate species diversity, and we also found a significant relationship between functional diversity and AGB within threshold values. Overall, the results indicated support for both tested "mass ratio hypothesis" and "niche complementarity hypothesis", which seemed that they were non-mutually exclusive.

Additionally, Rao's exerted the most significant influence on AGB, suggesting that the functional diversity index is a better indicator of biomass variation.

**Author Contributions:** Y.W. conceptualization, data curation, methodology, software, writing–original draft preparation, writing–reviewing and editing. S.W. conceptualization, writing–reviewing and editing. Y.L. conceptualization, methodology, investigation, writing–reviewing and editing, X.L. and J.Z. data curation, investigation, formal analysis. All authors have read and agreed to the published version of the manuscript.

**Funding:** National Natural Science Foundation of China (nos. 42101074, 51779241), General Program of Natural Science Foundation of Chongqing (no. cstc2020jcyj–zdxmX0018), The Three Gorges' follow-up scientific research project from Chongqing Municipal Bureau of Water Resources (no. 5000002021BF40001) and Venture and Innovation Support Program for Chongqing Overseas Returnees (no. CX2019023) supported this work.

**Institutional Review Board Statement:** Not applicable.

**Informed Consent Statement:** Not applicable.

**Data Availability Statement:** Not applicable.

**Conflicts of Interest:** The authors declare no conflict of interest.

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
