# Peer review of "Functional Trait Diversity Shapes the Biomass in the Dam-Induced Riparian Zone"

_agriculture, doi:10.3390/agriculture12030423_

Round 1

Reviewer 1 Report

The MS is much improved and only few and  minor changes are required. 

LL246-248/ indicate the type of vegetation formations (eg. forests, shruby vegetation, grasslands, ...).

Figure 1. Add in the title of figure  "Elevation map or DEM".

LL253-254: add the amount of rainfall during September of 2019 and 2020.

Throughout the figures, whenever the GAM is used, add the % of explained deviance.

Author Response

Major points: (“C” means Comment, “R” means Response)

Reviewer #1

C1: L246-248 indicate the type of vegetation formations (eg. forests, shruby vegetation, grasslands, ...).

R1:Thank you for your kind review. We added the type of vegetation in 2.1 Study area. “The type of vegetation is herbaceous, which is dominated by Cynodon dactylon, Xanthium sibiricum, Cyperus rotundus, Alternanthera philoxeroides and Setaria viridis”. Please see Line 105-107.

C2: Figure 1. Add in the title of figure "Elevation map or DEM".

R2: Many thanks for your comments. We have changed the title “The location of sampling sites in the WFLZ of the TGR” to “The location of sampling sites and DEM in the WFLZ of the TGR”. Please see Lines 110.

C3: LL253-254: add the amount of rainfall during September of 2019 and 2020.

R3:Many thanks for your comments. We have added the contents in 2.1 Study area “The amount of rainfall in study area is around 150-170 mm during September of 2019 and 2020” .Please see Line 103-104.

C4: Throughout the figures, whenever the GAM is used, add the % of explained deviance.

R4: Thanks for your suggestions. We have checked the whole manuscript and made sure that we have added the % of explained deviance.

Reviewer 2 Report

Dear Authors,

the paper has been highly improved

Author Response

Thanks a lot!

This manuscript is a resubmission of an earlier submission. The following is a list of the peer review reports and author responses from that submission.

Round 1

Reviewer 1 Report

Comments for the Author:

Review on” Functional trait diversity shapes the biomass in the dam-induced riparian zone” provide us with results of impact of dames on riparian plant communities. Generally, I think that the topic of the paper is very interesting and brings new original data for this topic.  I recommend a minor revision.

Further comments:

Abstract:

Line 14: Replace the word “Yet” maybe with “In advance”

Line 18: Delete “By our investigation” or replace.

Keywords:

Delete keywords that are already in the title. It's a duplication.

Methods:

Line 110 – 112: I do not understand how you have in total 59 quadrats in you have 4 transect with 3-4 quadrats. In this calculation you have at most 16 quadrats. Please write more clearly.

Line113: Percent cover (0 – 100 % vertical projection) – please describe in more details.

Line 115 – 118: Authors have special subchapter of “Traits measuring”. The sentences from “Following the standard …”  write there. This will make it more transparent.

Line 119 – 128: On the beginning of this subchapter, I will expect list of traits that you used.

In the following, we systematically describe the methods. You now have duplication of information.

Line 133: I am confused how you calculated CWM, you have only percent cover of the species and here you write “number of species per plot”?

Line143: Missing citation of the program.

Results:

In general: Subtitles from the Figures should be self-contained. Abbreviations must be described in the legend in Subtitles.

Line 157: The title should not begin with an abbreviation.

Reviewer 2 Report

Instead of only using GAM, I recommend to test the linear relationships using GLM. For the NL relationships that have a humped-shape, I suggest to fit the data to a parametric model and give in the results the equation of the relationship along with all the statistics.  

In Figures 1, 2, 3 and 4, precise to what test refers the P-values? give reference to letters a, b, c... if the figure legend? Explain in the legend of figure the meaning of the red curves? Add the explained deviance (%) of GAM for each relationship. 

Reviewer 3 Report

Dear Authors,

Presented paper is really interesting, but the part of study area should be improved. Here I prepose to get a map or some maps to show location of study areas.

The topic of manuscript ID-agriculture-1558658 is interestings and important for science. Today we have a lot atnthropogenic constructions as dumps, which could have not possitve impact on plants/vegetations.

but my main question is how they results could be use in practic aspects? How directions they could formulated for local gaverments, etc.